# Study on the Phase Angle Master Curve of the Polyurethane Mixture with Dense Gradation

**Haisheng Zhao** [1,2], **Xiufen Wang** [3], **Shiping Cui** [1], **Bin Jiang** [4], **Shijie Ma** [1,*], **Wensheng Zhang** [4], **Peiyu Zhang** [1], **Xiaoyan Wang** [1], **Jincheng Wei** [1] and **Shan Liu** [1]

1 Key Laboratory of Highway Maintain Technology Ministry of Communication, Jinan 250102, China; zhaohaisheng@sdjtky.cn (H.Z.); cuishiping@sdjtky.cn (S.C.); zhangpeiyu@sdjtky.cn (P.Z.); wangxiaoyan@sdjtky.cn (X.W.); weijincheng@sdjtky.cn (J.W.); liushan0378@163.com (S.L.)
2 School of Highway, Chang'an University, Xi'an 710064, China
3 Qingdao Highway Development Center, Qingdao 266075, China; w_xiufen@163.com
4 Wanhua Chemical Group Co., Ltd., Yantai 265599, China; 2017021062@chd.edu.cn (B.J.); wszhangb@whchem.com (W.Z.)
* Correspondence: mashijie@sdjtky.cn; Tel.: +86-186-6016-3082

**Abstract:** The phase angle master curve of the PU mixture is a new research field that is urgently needed to characterize the viscoelastic of the PU mixture under different conditions. In this paper, five master curve models, five shift factor equations, and four error minimization methods were introduced to fitting the phase angle master curve of the PU mixture. The results analysis indicated that the master curves fitted by different error minimization methods had small differences when the loading frequency was higher than $10^{-3}$ Hz. The $R^2$ maximization as the main constraint and the others as the additional constraints were recommended as the error minimization method. The combination of the Christensen Anderson and Marasteanu model (CAM) and kaelble shift factor equation was recommended for fitting the phase angle master curve of the PU mixture. The phase angle master curve of the PU mixture did not follow the "Bell" shape of the asphalt mixture. The PU mixture with smaller temperature susceptibility would still be subject to the PU at higher temperatures and was closer to that of the viscoelastic material. The phase angle master curve construction was analyzed for the first time and proper master curve fitting parameters were recommended for pavement performance predicting and analyzing.

**Keywords:** polyurethane mixture; phase angle; master curve model; shift factor equation; error minimization method; viscoelastic

## 1. Introduction

The asphalt pavement mechanics analysis method used in flexible pavement design is used as an effective method to predict the performance of the asphalt pavement, and this method presents many advantages, e.g., convenience, low analysis cost, and more factors that influence the pavement's performance [1,2]. The material characterization method and its accuracy [3] greatly influence the mechanistic analysis and pavement structure design of asphalt pavements. Therefore, before pavement designing, the pavement material characteristic and pavement load response should be studied and analyzed. The asphalt pavement material is a viscoelastic system; the asphalt mixtures manifest more complex linear viscoelastic (LVE) characteristics under small strain conditions (100–150 με) due to the combination of the viscoelastic asphalt binders and the aggregate skeleton [4–10]. It is necessary to understand the rheological property for analyzing the material response, material selection, and pavement design concerning calculating the required pavement thickness [8,11–15].

Various methods could be adopted to characterize the viscoelastic properties and predict the structural performance and dynamic response in pavement design [16]. In

the Mechanistic—Empirical Pavement Design Guide (MEPDG), the dynamic modulus is (1) adopted to characterize the viscoelastic property of the asphalt mixture and (2) determine the structural capacity of pavements; it is also (3) the most fundamental input parameter for pavement design and material property evaluation of asphalt pavement. The dynamic modulus test result could be used to (1) define the stiffness characteristic as a function of loading frequency and temperature [17]; (2) characterize the LVE behavior (temperature−time-dependent property) of the asphalt mixture under various temperatures and loading frequencies; (3) calculate the stress (or strain) response of the asphalt pavement structure under desired dynamic loading and climate conditions in the pavement structural design [18]; (4) predict the asphalt pavement performance [18]. Therefore, while designing flexible pavement and evaluating the pavement's structural performance, the phase angle is one of the most critical material properties [19,20].

The master curves of the dynamic modulus and phase angle can be built by employing the time−temperature superposition principle (TTSP) [21–26]. The reasons for constructing master curves are as follows [27]: (1) predicting the dynamic modulus and phase angle at temperatures and loading frequencies when test equipment or time is limited; (2) modeling the asphalt pavements under all possible climates and loading conditions; (3) comparing the performance of asphalt mixtures. The master curve provides the main dynamic parameters of viscoelastic materials, and the phase angle can better reflect the material's relaxation characteristics [28]. If the same dynamic modulus of the materials is the same, but the phase angles are different, this means that the materials are different [29]. The degree of viscous and elastic behavior of the materials at a particular temperature and frequency is referred to as the phase angle. As one of the principal measures of viscoelastic behavior, the phase angle reflects the phase delay between the amplitude of applied stress and corresponding strain [30]. In addition, phase angle (δ) could be used to divide the complex modulus (*E*\*) into its two inherent parts: storage and loss modulus. Therefore, determining the δ-value is essential for superior pavement analysis.

The experimental data of phase angle can be fitted into mathematical models [31] for the continuous master curve. During the past decades, a consistent number of studies have been conducted to develop master curve modeling solutions for describing the phase angle evolution for asphalt binder and mixture [6,8,11,12,14,32–34]. Those models could be used to predict the stiffness of a material at temperatures and frequencies that are beyond the experimental scope [35]. The concept of predicting phase angle from the slope of the complex modulus versus frequency, named the K−K relations, was first suggested by [36]. Study [37] established the function form for the phase angle master curve based on the K−K relations using the sigmoidal model and generalized sigmoidal model. Based on dynamic modulus data for asphalt mixtures, reference [3] generated the phase angles and assessed the accuracy of the predictions using the Black space diagram. Research [38] performed the aging test in the laboratory to investigate the effect of aging on the dynamic modulus and phase angle; phase angle master curves were successfully predicted over a range of ages, validating the K−K relations—based phase angle prediction model. Study [39] provided formulae that could be used to analyze master curves of both dynamic modulus and phase angle using a similar basis for sigmoidal forms. Works [36,39,40] demonstrated that the phase angle could be modeled using the sigmoidal function. The research performed in [37] proposed a sigmodal equation and regression parameters to determine the phase angle master curve from the dynamic modulus vs. frequency data.

The PU mixture is a new mixture for pavement engineering which possesses higher road performance. Many studies were performed to evaluate the improvement of the road performance of the PU mixture, the viscoelastic and rheology properties of the PU mixtures were studied by researchers [41], but the phase angle and the construction of the master curve of the PU mixture attract little attention. The phase angle is an important parameter to reflect the inherit viscoelastic property and the influence of higher temperature or lower loading frequency on the performance of the PU mixture; therefore, it is important

to study the phase angle and the construction of the master curve of the PU mixture before application.

In this paper, five different mathematical models, five shift factor equations, and four error minimization methods for fitting the phase angle master curve of the asphalt mixture were introduced to fit the tested phase angle of the PU mixture. Error analysis and goodness-of-fit statistics were used to evaluate the accuracy of the extrapolation calculations. Firstly, the fitting results under different error minimization methods were compared with the same prediction models and shift factor equation for determining the effect of the error minimization method on the accuracy of fitting results. Secondly, after determining the most effective error minimization method, the influence of different shift factor equations on the fitting accuracy for different prediction models was evaluated. Finally, the phase angle master curves under different models with the selected shift factor equations were graphically and statistically compared and analyzed. This paper aims to provide a simple recommendation on the selection of the proper master curve model and shift factor equation combination for the phase angle master curve construction of the PU mixture for a follow-up study.

## 2. Materials and Methods

### 2.1. Material

The gradation of the PU mixture used in this study was dense gradation, and the aggregate was basalt aggregate. The gradation and the binder content were determined by using the SBS-modified asphalt binder according to the Marshall method, then the PU binder replaced the SBS–modified asphalt with the same content to fabricate the specimens. According to the Marshall test results, the gradation was plotted, shown in Figure 1. The *x*–axis was in 0.45 power scale, and the *y*–axis was in arithmetic scale.

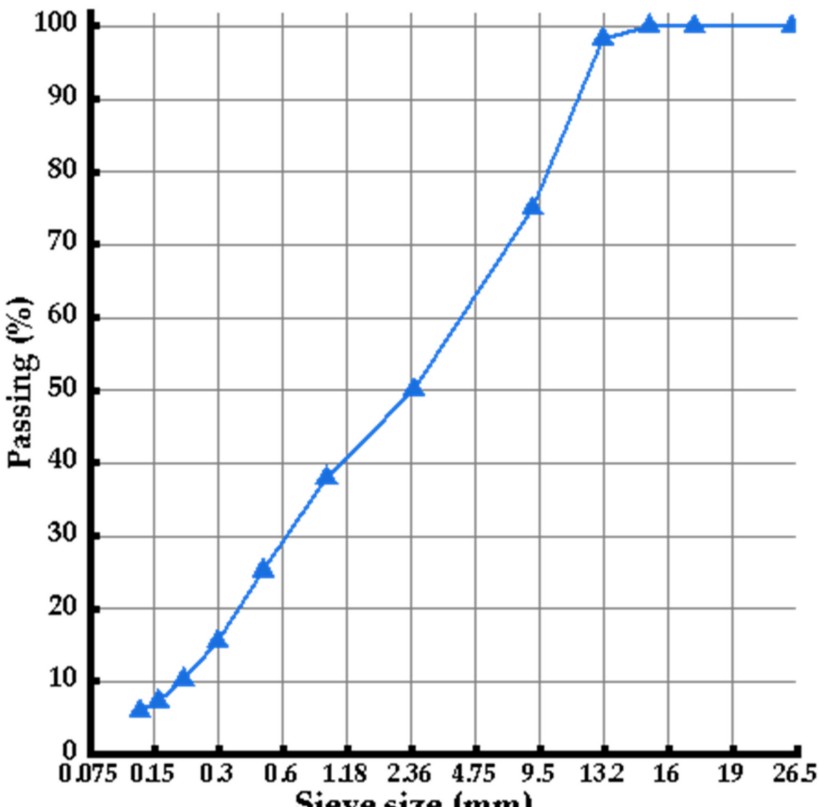

**Figure 1.** Gradation of PU mixture.

The optimum ratio of PU binder to aggregate was 5.3%, and the PU binder, which is a kind of single component PU, was supplied by Wanhua Chemical Group Co., Ltd. (Yantai, China). The PU binder solidified after being cured in wet conditions.

The specimen fabricating and dynamic modulus test both followed the procedure and requirements in the literature [42]. The specimens were compacted 100 times by the Superpave gyratory compactor (SGC) (Pine Test Equipment, Inc., Grove City, PA, USA); after compacting, the specimens were cored and sawed into the dimension of 150 mm in height and 100 mm in diameter. The test was performed on the Asphalt Mixture Performance Tester (AMPT, Pine Test Equipment, Inc., Grove City, OH, USA) based on AASHTO: TP–62 (2009); the strain control mode and the sinusoidal loading waveform were adopted in the test. A strain limitation of 75–125 με was used during the test procedure to keep the specimens within the elastic range. The phase angle of the PU mixture was measured in the experiment at six temperatures (5, 15, 25, 35, 45, 55 °C) and eight loading frequencies (0.1, 0.5, 1, 2, 5, 10, 20, 25 Hz). The average phase angle values of four replicates were used for fitting the master curve.

## 2.2. Methodology

By horizontally shifting data acquired at multiple temperatures and frequencies to build a single smooth and continuous master curve at the arbitrary reference temperature, the phase angle data may be utilized to construct the master curve, following the theoretical foundation of TTSP. The models used for constructing the phase angle master curve also had related equations that could be used to construct the phase angle master curve. In this paper, five models were used to build the phase angle master curve of the PU mixture, and the reference temperature in this study was set as 20 °C.

### 2.2.1. Phase Angle Fitting Model

According to the findings of research [36], the relationship of complex modulus against frequency as shown in Equation (1), which is expressed in terms of the approximate K−K relations [43,44], could be used to establish the phase angle master curve's function form.

$$\delta(f_r) = \frac{\pi}{2} \left[ \frac{d \log|E^*(u)|}{d \log(u)} \right]_{u=f_r}, \tag{1}$$

where $u$ is the integral variable.

The phase angle master curve could describe the elastic and viscous characteristics of the asphalt mixture. In this paper, five models were adopted from the corresponding dynamic modulus master curve equations and shown as follows:

### Model 1: Standard Logistic Sigmoid Model

The Standard logistic Sigmoid model (SLS) of phase angle [45] can be seen in Equation (2).

$$\delta(f_r) = -\frac{\pi}{2} \times \frac{\alpha \times \gamma \times e^{\beta + \gamma \times \log(f_r)}}{\left(1 + e^{\beta + \gamma \times \log(f_r)}\right)^2}, \tag{2}$$

where $\delta(f_r)$ is the phase angle in °; $f_r$ is the load frequency at the reference temperature in Hz; $\alpha$, $\beta$, and $\gamma$ are the fitting parameters. $\beta$ and $\gamma$ are the shape parameters of the model curve, which describe the shape of model 1 as depicted in Figure 2. In Figures 2–12 and 20, the $x$-axis represents the loading frequency in logarithm form, the $y$-axis represents the phase angle in arithmetic form.

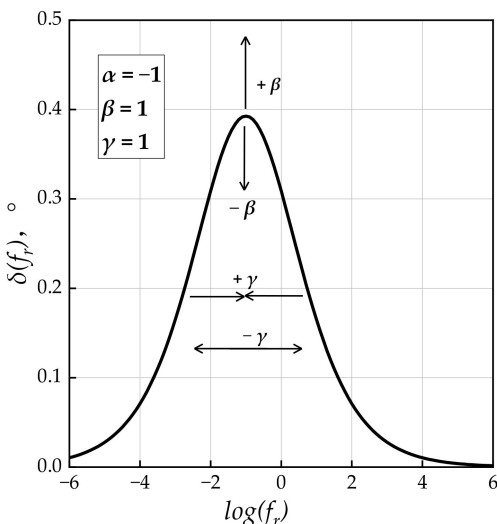

**Figure 2.** Graphical interpretation of Model 1.

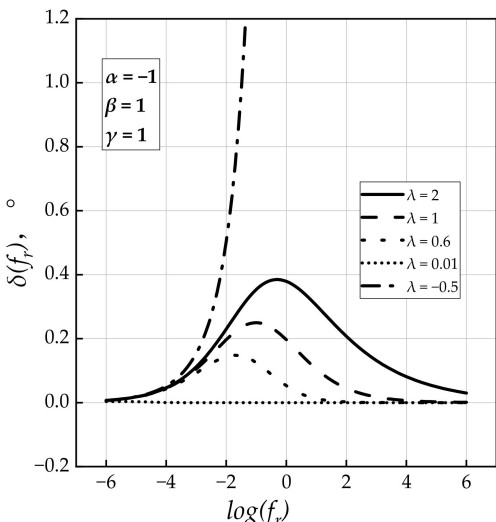

**Figure 3.** Graphical interpretation of Model 2.

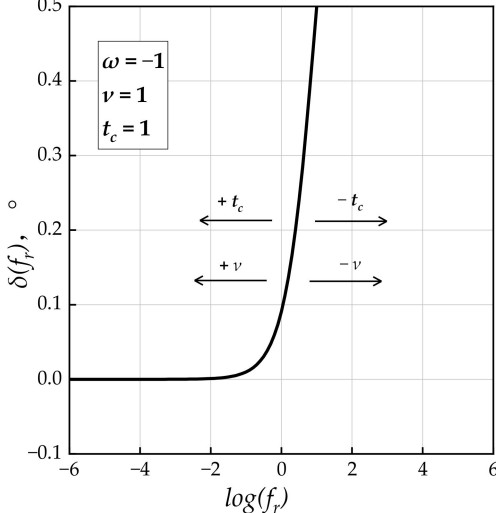

**Figure 4.** Graphical interpretation of Model 3.

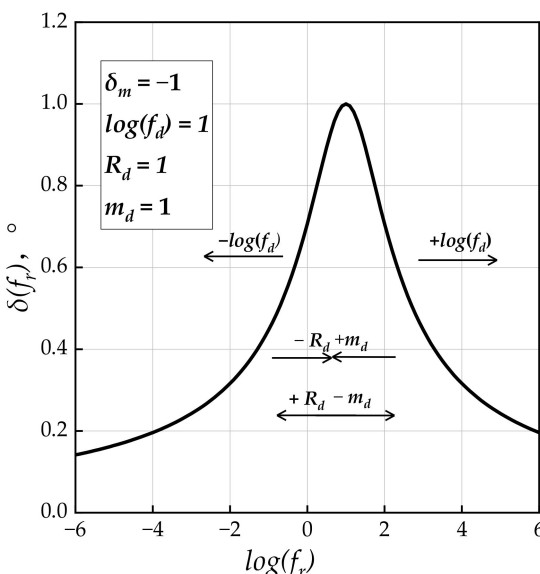

**Figure 5.** Graphical interpretation of Model 4.

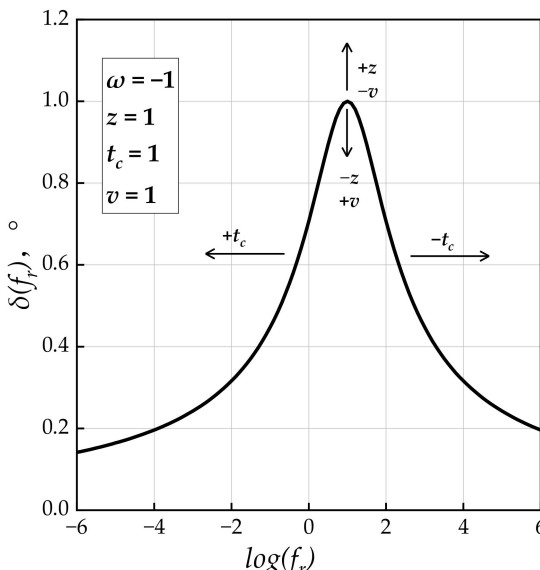

**Figure 6.** Graphical interpretation of Model 5.

Model 2: Generalized Logistic Sigmoidal (GLS) Model

The GLS model, which was introduced by [46], could represent and fit the asymmetric evolution of phase angle data. Based on the K−K relations, phase angle can be easily computed based on [36] as can be seen in Equation (3):

$$\delta(f_r) = -\frac{\alpha \times \gamma \times e^{\beta + \gamma \times log(f_r)}}{\left(1 + e^{\beta + \gamma \times log(f_r)}\right)^{\frac{1}{\lambda} + 1}},\tag{3}$$

where $\alpha$, $\beta$, and $\gamma$ are the fitting parameters; $\lambda$ is the additional fitting parameter which is used to depict the asymmetrical shape of the function. $\beta$ and $\gamma$ describe the shape of model 2 as depicted in Figure 3.

Model 3: Christensen Anderson and Marasteanu Model (1999)

Model 3 for phase angle expressions is given by the following Equation (4). Note that the equation of the phase angle [36] was calculated using the same Booji and Thoone approximation that was presented in the previous section.

$$\delta(f_r) = -\frac{w \times \left(10^{log(f_r)-t_c}\right)^v}{1 + \left(10^{log(f_r)-t_c}\right)^v},$$

(4)

where $v$, $w$, and $t_c$ are the fitting parameters and $v$, and $t_c$ could describe the shape of Model 3 as depicted in Figure 4.

It is also worth noting that the sigmoidal structure is absent from the mathematical expression of Model 3 [47,48]. The exponential-based function shown by Equations (2) and (3) present a smooth bell-shaped curve when used to fit data. However, as can be seen in Equation (4), Model 3 provides [48] a combination of logarithm and exponential functions that deviates from a bell-shaped pattern. Therefore, this means that the phase angle master curves may be remarkably different from the ones generated by Models 1 and 2.

Model 4: Modified CAM Model (2001)

To improve the compatibility with experimental data on the phase angle, Zeng and his co-authors proposed a modified version of Model 3 [8]. An empirical mathematical expression for phase angle was introduced based on Equation (5) [8], which was provided with the model function of phase angle.

$$\delta(f_r) = \delta_m \times \left[1 + \left(\frac{log(f_d) - log(f_r)}{R_d}\right)^2\right]^{-\frac{m_d}{2}},$$

(5)

where $\delta_m$ is the phase angle constant at $f_d$; $f_d$ is the location parameter with a dimension of frequency in Hz; $R_d$ and $m_d$ are shape parameters, which could describe the shape of Model 4 as shown in Figure 5.

It needs to be mentioned that Model 4 does not take into consideration the straightforward relationship between the phase angle and the corresponding phase angle, as shown in [36]. This model's fundamental drawback is that the equation for the phase angle relies on an empirical approximation because of the varying values in Equation (5). However, a better fitting of the experimental results was observed compared to Model 3 in different studies [8,48], likely as a consequence of the increased number of parameters.

Model 5: Sigmoidal CAM Model (SCM Model)

Given the mathematical limitations of Model 3, this paper introduced a further modified version of the latter [48], which combines the CAM model's simplicity and reasonable physical meaning (e.g., $w$ means the velocity of phase angle and $v$ represents relaxation spectrum) with the benefit of the sigmoidal function. Based on Equation (6) [36], the model is named the Sigmoidal CAM Model (SCM) of phase angle, and can be expressed as follows:

$$\delta(f_r) = -\frac{w \times z \times e^{z \times log(f) + t_c}}{v^2 \times \left(1 + e^{z \times log(f) + t_c}\right)^{1 + \frac{1}{v}}},$$

(6)

where $v$, $w$, and $t_c$ are consistent with the parameters of Model 3, and $z$ is the newly introduced fitting parameter. The parameters $v$, $z$, $w$, and $t_c$ related to the shape of Model 5 are depicted in Figure 6.

2.2.2. Shift Factor Equation

To generate a phase angle master curve, data obtained at different temperatures and frequencies must be related to each other to form a unique curve of material stiffness representing the stiffness of the material. The shift factor was temperature-dependent,

reflecting the temperature-specific translation of the modulus curve used to generate the overall master curve. The phase angle changes as a function of temperature, and its general form is given by Equation (7). By using the TTSP while modeling the master curve, the phase angle at varying test temperatures may be shifted to the lower frequency of the master curve. The $f_r$ value is the frequency equivalent of the experimental temperature relative to the reference value. Once the shift factor is known, Equation (7) could be used to calculate the new, lower frequency.

$$f_r = f \times \alpha_T, \tag{7}$$

$$log(f_r) = log(f) + log(\alpha_T), \tag{8}$$

where $log(f)$ is the frequency in experiment temperature; $log(f_r)$ is the reduced frequency in reference temperature.

Five commonly used shift factor equations were employed in this research, includingthe Log-linear equation, Polynomial equation, Arrhenius equation, WLF equation, and Kaelble equation.

Equation (1): Log-Linear Equation

The use of the log-linear equation is one of the most popular temperature-shifting methods for asphalt mixtures. For many binders, $log(\alpha_T)$ is found to vary linearly with temperature below 0 °C, as stated by Christensen and Anderson [40], and this similar relationship has been regarded appropriate for asphalt mixture at low to moderate temperatures [49]. The log-linear equation for calculating the shift factor is

$$log(\alpha_T) = C \times (T - T_r), \tag{9}$$

where $log(\alpha_T)$ is the shift factor, $T$ is the temperature in °C, $T_r$ is the reference temperature (20 °C), and the constant $C$ is calculated from the results of the experiment.

Equation (2): Polynomial Equation

The Polynomial equation [34,50] may be stated as follows and is shown to be a good fit for the shift factors across a board temperature range:

$$log(\alpha_T) = a \times (T - T_r) + b \times (T - T_r)^2, \tag{10}$$

where $a$ and $b$ are the parameters of regression.

Equation (3): Arrhenius Equation

Equation (11) shows the Arrhenius equation [51] which is used for calculating the shift factor:

$$log(\alpha_T) = \frac{E_a}{19.147142} \times \left( \frac{1}{T + 273.15} - \frac{1}{T_r + 273.15} \right), \tag{11}$$

where $C$ is a constant, and the Arrhenius equation which can describe the behavior of the material below $T_g$ [52] has only one constant that needs figuring out.

Equation (4): Williams−Landel−Ferry Equation

To evaluate the shift factor of the asphalt mixture, the Williams−Landel−Ferry (WLF) equation [53,54] is often employed to describe the relationship between the shift factor and temperature above $T_g$:

$$log(\alpha_T) = \frac{-C_1 \times (T - T_r)}{C_2 + (T - T_r)}, \tag{12}$$

where $C_1$ and $C_2$ are the two parameters of regression.

Equation (5): Kaelble Equation

Equation (13) illustrates that the Kaelble equation, a modified version of the WLF equation, may be employed to describe the shift factor−temperature relationship below $T_g$,

$$log(\alpha_T) = \frac{-C_1 \cdot (T - T_r)}{C_2 + |T - T_r|}, \tag{13}$$

where $C_1'$ and $C_2'$ are the two regression parameters.

### 2.3. Error Minimization Method

Minimizing the sum of the error between the predicted data and measured data is the goal of regressing the parameters of the aforementioned models. To construct the master curves deriving from the experimental data, the nonlinear least squares regression analysis was integrated into the Microsoft Excel Solver's error minimization procedure. Due to cases fitting well, the constraint range of the variables was not defined. The Generalized Reduced Gradient (GRG) Nonlinear was chosen as the method for solving the problem. Furthermore, each fitting procedure began with identical initial settings for fitting parameters. The variables were optimized such that the phase angle master curve had maximum overlap with the measured results.

To evaluate five phase angle model's fitting divergence and minimize the error between the predicted and measured phase angle, four different error minimization methods were applied to determine the "goodness of fit" [55]: (1) Se/Sy minimization, as in Equations (14) and (15), (2) $R^2$ maximization as in Equation (16), (3) the Sum of Square Error (SSE) minimization as in Equation (17), and (4) Error$^2$ minimization as in Equation (18).

The following definitions [35] apply to both standard error of estimate and standard error of deviation:

$$Se = \sqrt{\frac{1}{(n - p - 1)} \times \sum_{i}^{n} |\hat{x}_i - x_i|^2}, \tag{14}$$

$$Sy = \sqrt{\frac{1}{(n - 1)} \times \sum_{i}^{n} |\hat{x}_i - \overline{x}_i|^2}, \tag{15}$$

where $x_i$ is the measured phase angle, $\hat{x}_i$ is the predicted phase angle, and $\overline{x}_i$ is the mean value of the measured phase angle.

To compute the error, the $R^2$ of a given model may be calculated using Equation (16):

$$R^2 = 1 - \frac{(n - p - 1) \times Se^2}{(n - 1) \times Sy^2}, \tag{16}$$

where $n$ is the sample size and $p$ is the number of parameters to be estimated.

The Sum of Square Error (SSE) between measured values after shifting $\delta_{measured}$ and predicted values $\delta_{predicted}$ is shown in Equation (18),

$$SSE = \sum \frac{\left(\delta_{measured} - \delta_{predicted}\right)^2}{\left(\delta_{measured}\right)^2}, \tag{17}$$

where $\delta_{measured}$ is the experimentally measured phase angle in $^\circ$ and $\delta_{predicted}$ is the predicted phase angle predicted by different master curve models and shift factor equations in $^\circ$. The *SSE* parameter could represent the relative error between the predicted and experimental measured phase angle. The coefficients of the models and shift factor equations were fitted to minimize *SSE*, defining the optimal results of the master curves.

A simultaneous minimization approach can be performed to fit the experimental results as can be seen in Equation (18),

$$Error^2 = \sum_{i=1}^{n} \left[\delta_{measured} - \delta_{predicted}\right]^2, \tag{18}$$

where the parameters correspond to those in the previous equation. The *Error²* value is the absolute difference between the predicted and the experimentally measured phase angle.

### 2.4. Fitting Process

The fitting process is shown as follows: (a) determining the effect of four error minimization methods on the fitting results. Under different phase angle master curve models, the shift equation factors $log(\alpha_T)$ in different shift factor equations were incorporated into the master curve model equation by Equation (9); then, the error minimization methods were separately used as the main constraint for fitting the measured phase angle data and minimizing the error between the predicted and measured phase angle data; the other three error minimization methods were used as the additional constraints. The difference between the phase angle master curves fitted by different error minimization methods was compared and analyzed to determine the more accurate error minimization method for each master curve model. (b) After determining the optimum error minimization method, the effect of the shift factor equation on the fitting results was analyzed by comparing the predicted and measured phase angle, and the most accurate shift factor equation was recommended for each master curve model. (c) Under the determined optimum shift factor equation, the difference between all phase angle master curve models was compared and analyzed to recommend the most accuracy model and shift factor equation for fitting the phase angle data.

## 3. Results

### 3.1. Analyzing the Influence of the Solver Method on the Fitting of the Phase Angle Master Curve

3.1.1. Model 1 Fitting Results

The phase angle master curve fitting results of Model 1 combined with five different shift factor equations under four error minimization methods are plotted in Figure 7. For further comparison, the master curves under the same shift factor equation are plotted in a single picture. In each picture, the phase angle master curve constructed by different error minimization methods is plotted together to compare the effect of the error minimization method on the fitting results of the phase angle master curve.

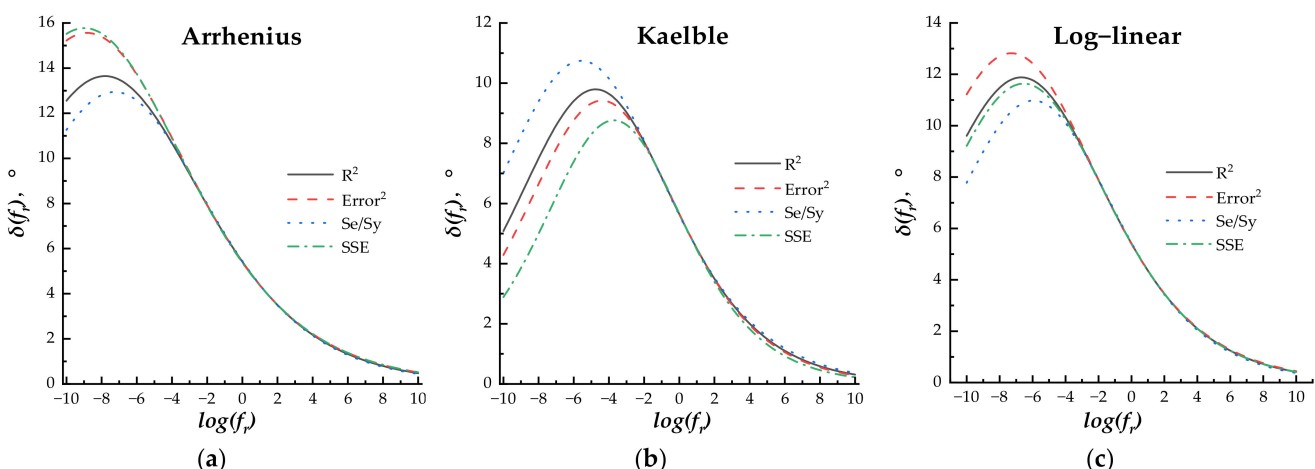

**Figure 7.** *Cont.*

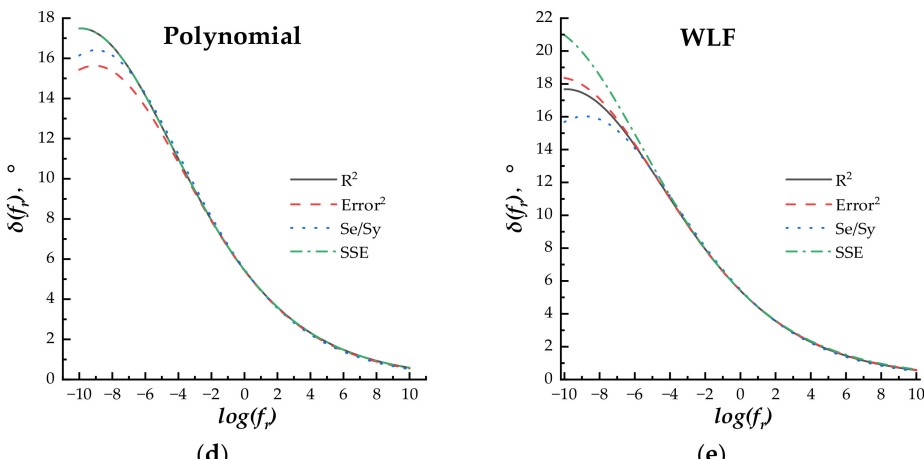

**Figure 7.** The master curve of Model 1 fitting results with different error minimization methods. (**a**) Arrhenius equation; (**b**) Kaelble equation; (**c**) Log-linear equation; (**d**) Polynomial equation; (**e**) WLF equation.

### 3.1.2. Model 2 Fitting Results

The phase angle master curve fitting results of Model 2 combined with five different shift factor equations under four error minimization methods are plotted in Figure 8. In each picture, the phase angle master curves of each shift factor equation are combined to compare the influence of the error minimization method on the fitting results.

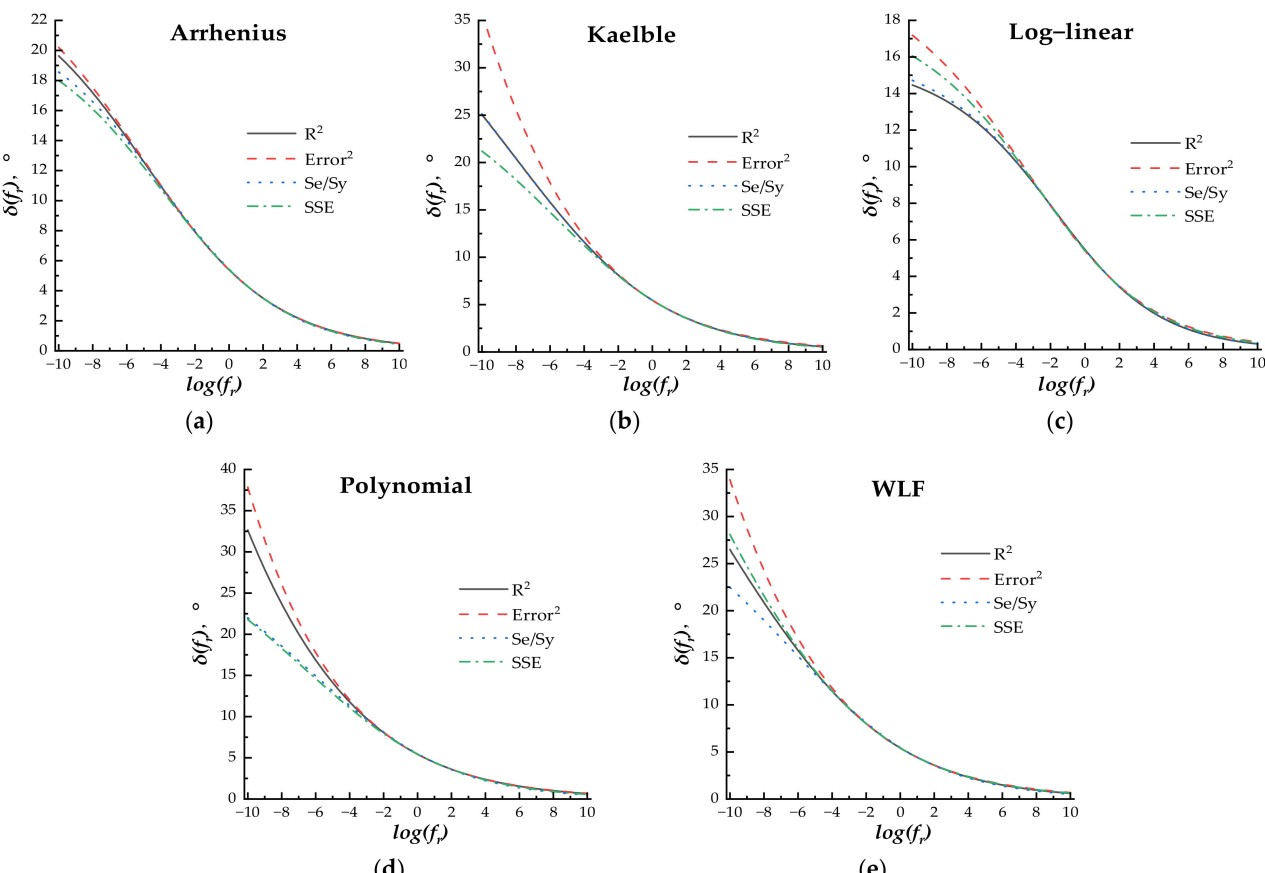

**Figure 8.** The master curve of Model 2 fitting results with different shift factor equations. (**a**) Arrhenius equation; (**b**) Kaelble equation; (**c**) Log-linear equation; (**d**) Polynomial equation; (**e**) WLF equation.

### 3.1.3. Model 3 Fitting Results

The phase angle master curve fitting results of Model 3 combined with five different shift factor equations under four error minimization methods are plotted in Figure 9. The phase angle master curves under every shift factor equation are combined to compare the impact of the error minimization method on the prediction accuracy of the phase angle master curve.

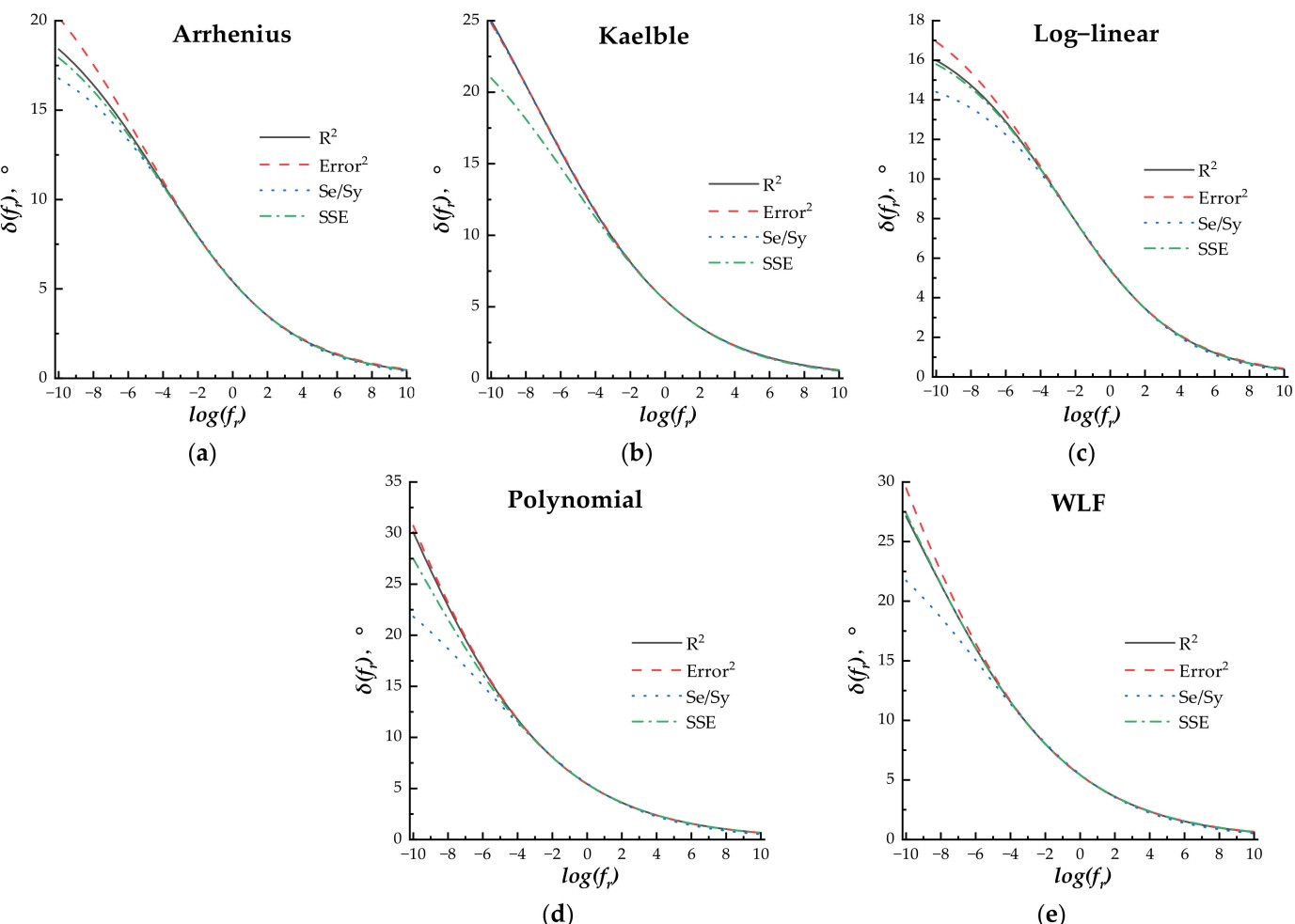

**Figure 9.** The master curve of Model 3 fitting results with different shift factor equations. (**a**) Arrhenius equation; (**b**) Kaelble equation; (**c**) Log-linear equation; (**d**) Polynomial equation; (**e**) WLF equation.

### 3.1.4. Model 4 Fitting Results

The phase angle master curve fitting results of Model 4 combined with five different shift factor equations under four error minimization methods are plotted in Figure 10. Different shift factor equations build different phase angle master curves; the master curves constructed by four error minimization methods under the same shift factor equation are plotted in an isolated picture to analyze the effect of the error minimization method.

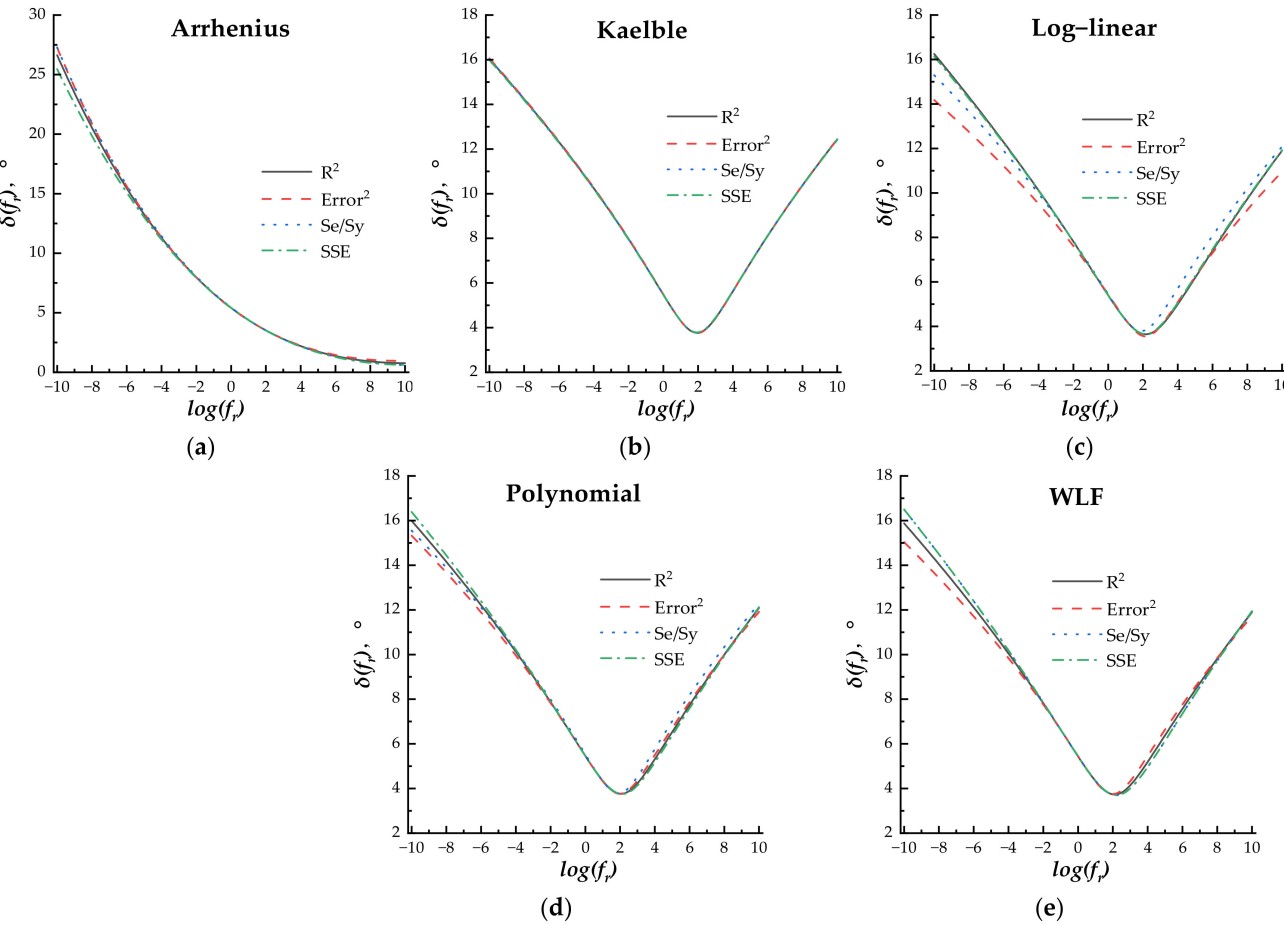

**Figure 10.** The master curve of Model 4 fitting results with different shift factor equations. (**a**) Arrhenius equation; (**b**) Kaelble equation; (**c**) Log-linear equation; (**d**) Polynomial equation; (**e**) WLF equation.

### 3.1.5. Model 5 Fitting Results

The phase angle master curve fitting results of Model 5 combined with five different shift factor equations under four error minimization methods are plotted in Figure 11. To compare the impact of the error minimization method on the phase angle master curves, the master curves of different error minimization methods under the same shift factor equation are plotted in an isolated picture.

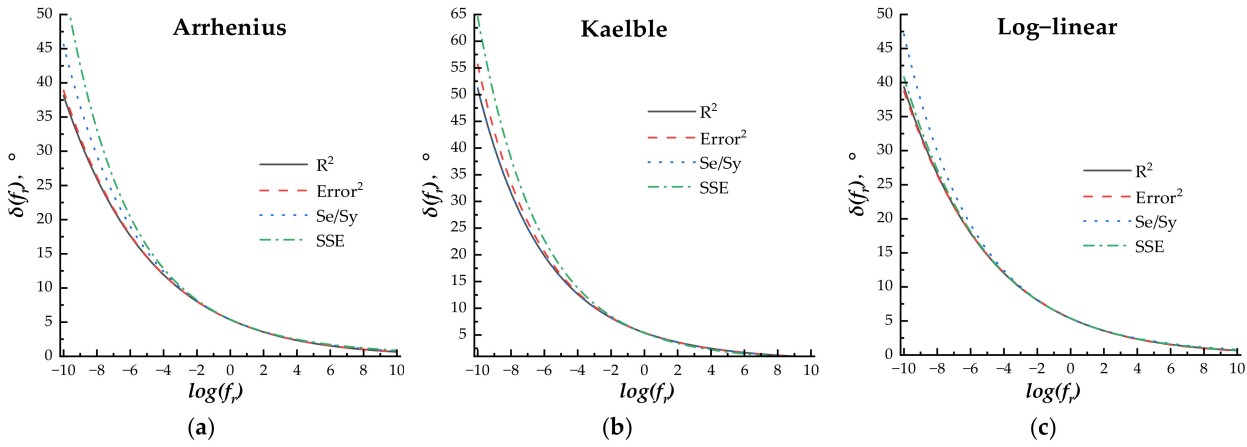

**Figure 11.** *Cont.*

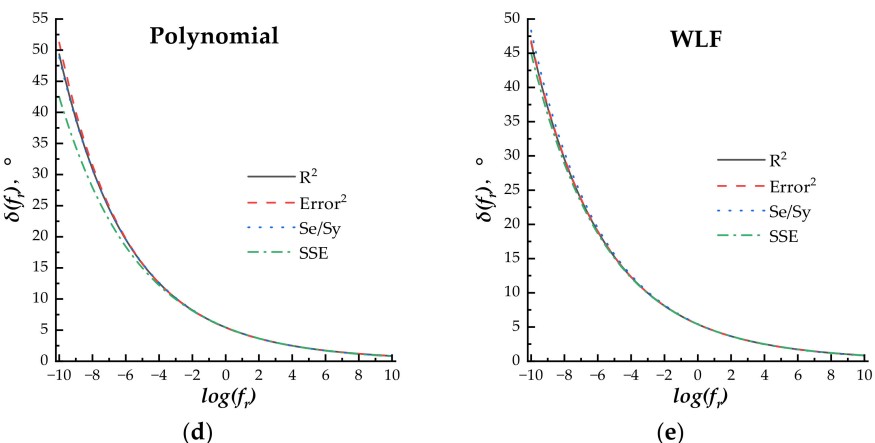

**Figure 11.** The master curve of Model 5 fitting results with different shift factor equations. (**a**) Arrhenius equation; (**b**) Kaelble equation; (**c**) Log-linear equation; (**d**) Polynomial equation; (**e**) WLF equation.

### 3.1.6. Comparison of Different Shift Factor Equations

For five different models, the phase angle master curves with different shift factor equations are plotted in Figure 12. In every single picture, the phase angle master curves constructed by different shift factor equations under the same model are shown and used to analyze the impact of the shift factor equation on the fitting results of the master curve under the same model. All the master curves were constructed by the $R^2$ maximization error minimization method as the main constraint.

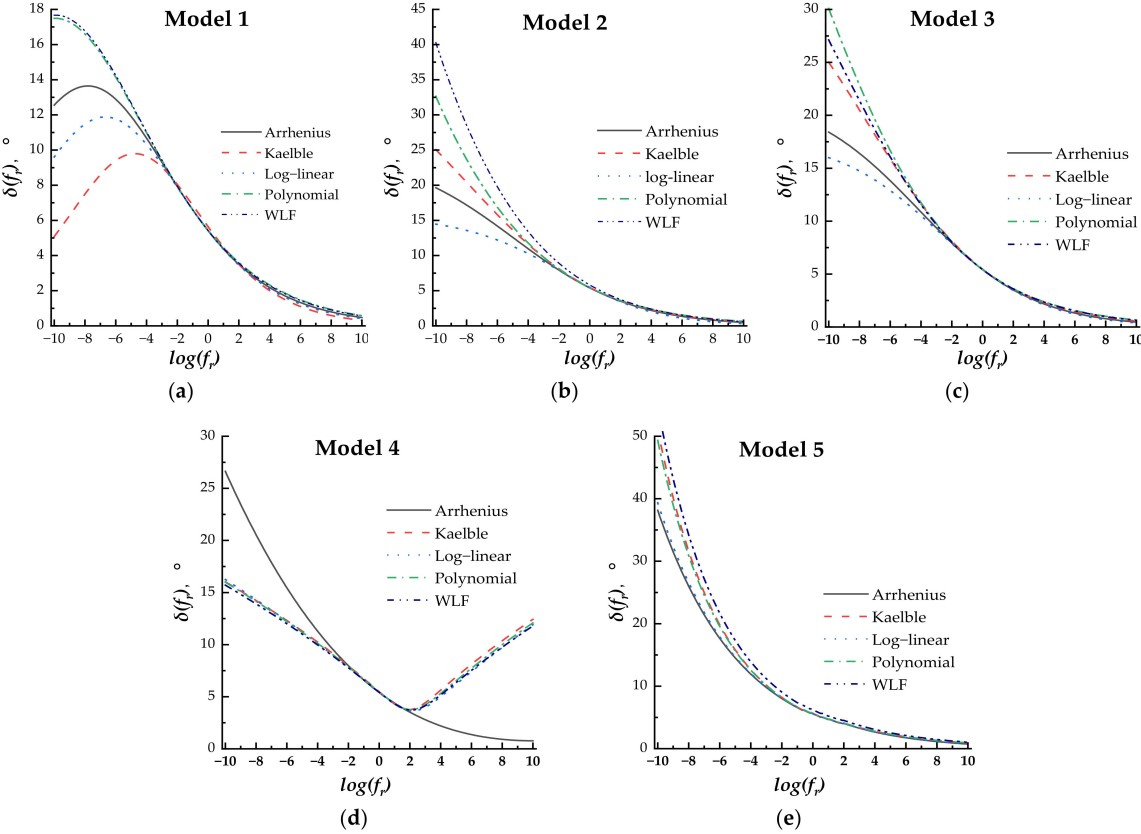

**Figure 12.** The master curve of different model fitting results with the $R^2$ minimization error minimization method. (**a**) the model 1 fitting result; (**b**) the model 2 fitting result; (**c**) the model 3 fitting result; (**d**) the model 4 fitting result; (**e**) the model 5 fitting result.

### 3.2. Analyzing the Influence of Shift Factor Equation on the Fitting of Phase Angle Master Curve

3.2.1. Model 1 Fitting Results

For comparison purposes, the measured data under all test temperatures and loading frequency and predicted phase angle values fitted by Model 1 for different shift factor equations are plotted in Figure 13. The linear fitting and line of equality (LOE) method were used to analyze the prediction accuracy. The linear fitting results of different shift factor equations are listed in Table 1. The residual sum of squares (RSS) represents the model fit degree between the measured and predicted phase angle data, which is the square of the distance between the real location of the spot and the location in the regressed line.

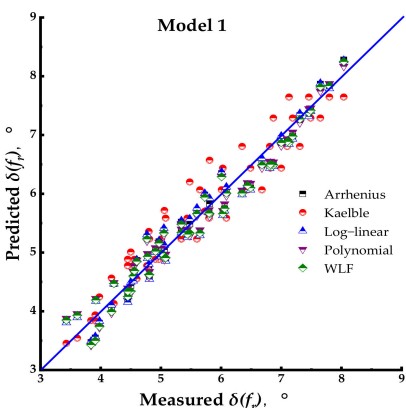

**Figure 13.** Comparison of predicted and measured phase angle for different shift factor equations under model 1.

**Table 1.** The linear fitting results of different shift factor equations under Model 1.

| Equation | Fitting Equation | $R^2$ | RSS |
|---|---|---|---|
| Arrhenius | $Y = 0.95353 \times X + 0.22017$ | 0.96271 | 2.77061 |
| Kaelble | $Y = 0.90081 \times X + 0.61604$ | 0.92021 | 5.53531 |
| Log-linear | $Y = 0.94968 \times X + 0.24469$ | 0.95881 | 3.04802 |
| Polynomial | $Y = 0.94916 \times X + 0.24175$ | 0.96584 | 2.50653 |
| WLF | $Y = 0.95157 \times X + 0.20955$ | 0.96376 | 2.67867 |

3.2.2. Model 2 Fitting Results

Figure 14 shows the measured phase angle under all temperatures and loading frequencies versus the predicted data fitted by Model 2 for different shift factor equations. To better visualize the prediction precision, the linear fitting results between the measured and predicted phase angle data of different shift factor equations are listed in Table 2.

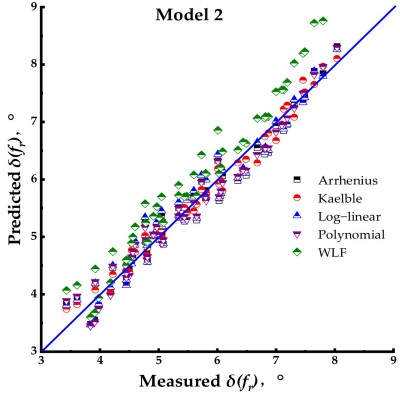

**Figure 14.** Comparison of predicted and measured phase angle for different shift factor equations under model 2.

**Table 2.** The linear fitting results of different shift factor equations under Model 2.

| Equation | Fitting Equation | $R^2$ | RSS |
|----------|------------------|-------|-----|
| Arrhenius | $Y = 0.95353 \times X + 0.22017$ | 0.96271 | 2.77061 |
| Kaelble | $Y = 0.90081 \times X + 0.61604$ | 0.92021 | 5.53531 |
| Log-linear | $Y = 0.94968 \times X + 0.24469$ | 0.95881 | 3.04802 |
| Polynomial | $Y = 0.94916 \times X + 0.24175$ | 0.96584 | 2.50653 |
| WLF | $Y = 0.95157 \times X + 0.20955$ | 0.96376 | 2.67867 |

### 3.2.3. Model 3 Fitting Results

Figure 15 compares the measured data under all test temperatures and loading frequencies and predicts the phase angle fitted by Model 3 for different shift factor equations. The linear fitting and LOE methods were adopted to compare the prediction accuracy, and the linear fitting results of all shift factor equations between the measured and predicted phase angle data are listed in Table 3.

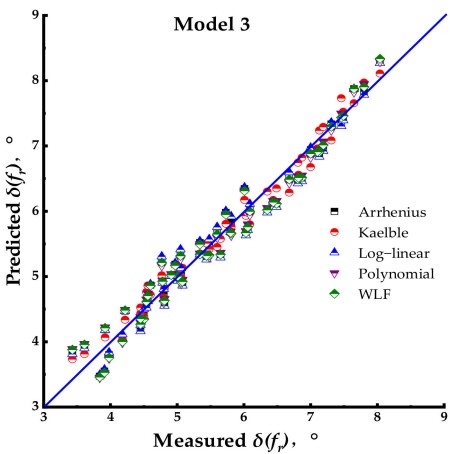

**Figure 15.** Comparison of predicted and measured phase angle for different shift factor equations under model 3.

**Table 3.** The linear fitting results of different shift factor equations under Model 3.

| Equation | Fitting Equation | $R^2$ | RSS |
|----------|------------------|-------|-----|
| Arrhenius | $Y = 0.95329 \times X + 0.22095$ | 0.96207 | 2.81832 |
| Kaelble | $Y = 0.9771 \times X + 0.11388$ | 0.9792 | 1.59565 |
| Log-linear | $Y = 0.94557 \times X + 0.26792$ | 0.95847 | 3.04782 |
| Polynomial | $Y = 0.95999 \times X + 0.18318$ | 0.96511 | 2.6209 |
| WLF | $Y = 0.95853 \times X + 0.18611$ | 0.96325 | 2.7579 |

### 3.2.4. Model 4 Fitting Results

Figure 16 gives the comparison between the measured and predicted phase angle fitted by Model 4 for different shift factor equations under all temperatures and loading frequencies. The linear fitting and LOE methods were introduced to measure the prediction accuracy of every shift factor equation, and the fitting results were listed in Table 4.

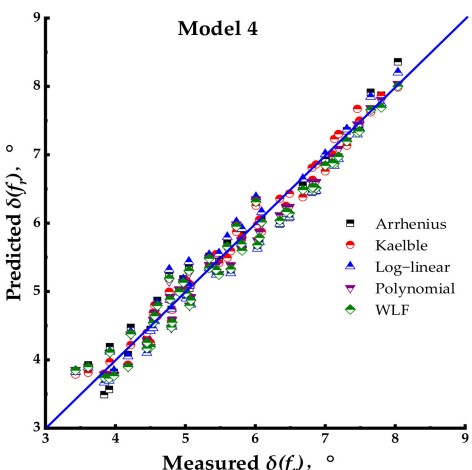

**Figure 16.** Comparison of predicted and measured phase angle for different shift factor equations under 4.

**Table 4.** The linear fitting results of different shift factor equations under Model 4.

| Equation | Fitting Equation | $R^2$ | RSS |
|---|---|---|---|
| Arrhenius | Y = 0.95517 × X + 0.21258 | 0.96108 | 2.90655 |
| Kaelble | Y = 0.98378 × X + 0.07795 | 0.98577 | 1.09921 |
| Log-linear | Y = 0.95141 × X + 0.22902 | 0.96121 | 2.87395 |
| Polynomial | Y = 0.95584 × X + 0.19652 | 0.97464 | 1.87013 |
| WLF | Y = 0.93769 × X + 0.26129 | 0.97083 | 2.07863 |

### 3.2.5. Model 5 Fitting Results

The comparison of the measured phase angle under all test temperatures and loading frequencies was plotted against the predicted phase angle fitted by Model 5 for different shift factor equations in Figure 17. The linear fitting and LOE methods were introduced to determine the prediction accuracy of different shift factor equations, and the fitting results of every shift factor equation are listed in Table 5.

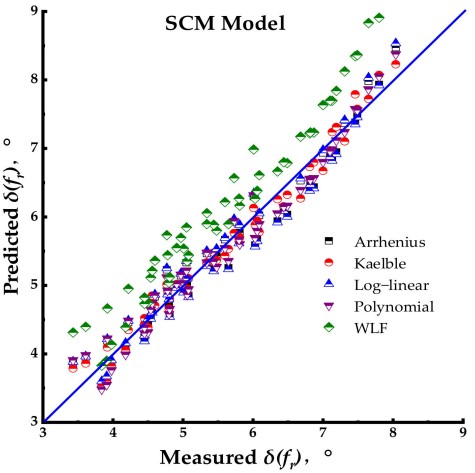

**Figure 17.** Comparison of predicted and measured phase angle for different shift factor equations under model 5.

**Table 5.** The linear fitting results of different shift factor equations under Model 5.

| Equation | Fitting Equation | $R^2$ | RSS |
|---|---|---|---|
| Arrhenius | $Y = 0.95939 \times X + 0.17299$ | 0.95809 | 3.1671 |
| Kaelble | $Y = 0.98029 \times X + 0.09528$ | 0.97729 | 1.75707 |
| Log-linear | $Y = 0.95075 \times X + 0.23697$ | 0.95163 | 3.61417 |
| Polynomial | $Y = 0.97043 \times X + 0.12692$ | 0.96435 | 2.73905 |
| WLF | $Y = 1.0891 \times X + 0.05371$ | 0.96013 | 3.87499 |

*3.3. Comparing Master Curves with Recommended Models and Shift Factor Equations*

3.3.1. Comparing Polynomial Shift Factor Equation Fitting Results

In this section, the predicted phase angle of different master curve models with the Polynomial shift factor equation was compared to the accuracy of prediction with the measured phase angle in Figure 18. The prediction statistics are organized in Table 6. According to the analyzed results, the models with the highest prediction accuracy would be recommended.

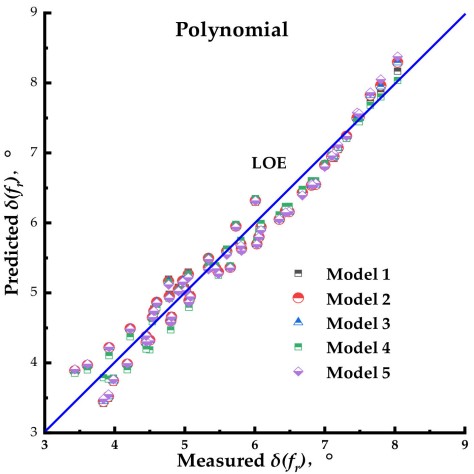

**Figure 18.** Comparison of predicted and measured phase angle with different master curve models (Polynomial shift factor equation).

**Table 6.** The linear fitting results of different master curve models with the Polynomial shift factor equation.

| Model | Fitting Equation | $R^2$ | RSS |
|---|---|---|---|
| Model 1 | $Y = 0.94916 \times X + 0.24175$ | 0.96584 | 2.50653 |
| Model 2 | $Y = 0.96169 \times X + 0.17361$ | 0.96506 | 2.63534 |
| Model 3 | $Y = 0.95999 \times X + 0.18318$ | 0.96511 | 2.6209 |
| Model 4 | $Y = 0.95584 \times X + 0.19652$ | 0.97464 | 1.87013 |
| Model 5 | $Y = 0.97043 \times X + 0.12692$ | 0.96435 | 2.73905 |

3.3.2. Comparing Kaelble Shift Factor Equation Fitting Results

In this section, the predicted phase angle of different master curve models with the Kaelble shift factor equation was compared with the measured phase angle in Figure 19. The prediction statistics are organized in Table 7. The models which had the highest prediction accuracy would be recommended in the following section.

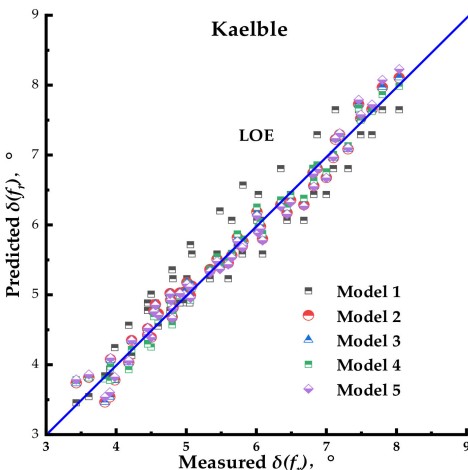

**Figure 19.** Comparison of predicted and measured phase angle with different master curve models (Kaelble shift factor equation).

**Table 7.** The linear fitting results of different master curve models with the Kaelble shift factor equation.

| Model | Fitting Equation | $R^2$ | RSS |
|---|---|---|---|
| Model 1 | $Y = 0.90081 \times X + 0.61604$ | 0.92021 | 5.53531 |
| Model 2 | $Y = 0.97563 \times X + 0.12184$ | 0.9789 | 1.61379 |
| Model 3 | $Y = 0.9771 \times X + 0.11388$ | 0.9792 | 1.59565 |
| Model 4 | $Y = 0.98378 \times X + 0.07795$ | 0.98577 | 1.09921 |
| Model 5 | $Y = 0.98029 \times X + 0.09528$ | 0.97729 | 1.75707 |

*3.4. Comparing the Master Curves under Different Models*

According to the discussion above, the master curves for modes with the Polynomial shift factor equation and the other four models with the Kaelble shift factor equation in a larger loading frequency range at the reference of 20 °C are plotted in Figure 20.

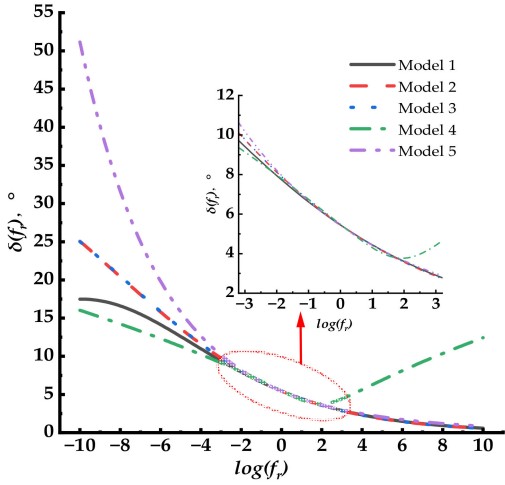

**Figure 20.** Master curve models of different models with WLF shift factor equation.

## 4. Discussion

*4.1. Analyzing the Influence of the Solver Method on the Fitting of the Phase Angle Master Curve*

4.1.1. Model 1 Fitting Results

The plots in Figure 7 indicate that the phase angle master curves fitted by different error minimization methods with the same shift factor equation had similar shapes; when

the loading frequencies were lower than $10^{-3}$ Hz, the curves had obvious differences. The master curves fitted by different shift factor equations showed different shapes, and the master curve peak values of different shift factor equations exhibited significant difference.

For all shift factor equations, when loading frequencies become larger than $10^{-3}$ Hz, the master curves under the same shift factor equation tend to be the same, and the differences are minimized. For the PU mixture used in road pavement under different vehicles, the loading frequency below $10^{-3}$ Hz would be rare.

For Model 1, four different error minimization methods would produce the same master curves and have insignificant influence on the phase angle master curve when the loading frequency is higher than $10^{-3}$ Hz. Based on the four parameter values ($R^2$, Se/Sy, error$^2$, SSE) fitted by four error minimization methods, the $R^2$ maximization method would produce the highest $R^2$ value, and the lowest Se/Sy, error$^2$, SSE values, and the ultimate phase angle value at the lowest frequency were different according to different error minimization methods; the ultimate value produced by the $R^2$ maximization method located in the middle and did not show extreme high or low values. The $R^2$ maximization method is adopted to obtain the regression parameters of all equations for Model 1 in the following discussion.

### 4.1.2. Model 2 Fitting Results

From Figure 8, it can be seen that the phase angle master curves fitted by different error minimization methods under the same shift factor equation had a similar trend. The ultimate phase angle values under the lower loading frequency exhibited a big difference; when the loading frequency was higher than $10^{-3}$ Hz, the difference between different master curves became insignificant, and all the lines converged.

For Model 2, when the loading frequency increased, the difference between different phase angle master curves under the same shift factor equation became negligible and coincided when the loading frequency was higher than $10^{-3}$ Hz. Therefore, the error minimization methods had an insignificant influence on the phase angle master curve at higher loading frequency. By comparing the $R^2$, Se/Sy, error$^2$, and SSE values fitted by different error minimization methods, the $R^2$ maximization method had relatively lower Se/Sy, error$^2$, and SSE values, and higher $R^2$ value. The ultimate value of the $R^2$ maximization method did not show extremely higher or lower compared with the other methods. In the following discussion, the $R^2$ maximization method was used for analyzing the phase angle master curve.

### 4.1.3. Model 3 Fitting Results

Under the same shift factor equation, the phase angle master curves in Figure 9 showed the same trend, and the phase angle values decreased with the increasing loading frequency. However, with the increasing loading frequency, the phase angle master curves trend to meet; when the loading frequency became higher than $10^{-3}$ Hz, the phase angle master curves were confluent.

For Model 3, the four different error minimization methods affected the phase angle under lower loading frequency. When the loading frequency increased, the effect of the error minimization method diminished gradually. For Model 3, the acceptable fitting results were first obtained by the $R^2$ maximization method based on the identical initial values. Therefore, for Model 3, the $R^2$ maximization used as the main constraint was selected for regressing the parameters in the following discussion.

### 4.1.4. Model 4 Fitting Results

For Model 4 in Figure 10, under the Arrhenius shift factor equation, the phase angle decreased with the increase in loading frequency; for the remaining four shift factor equations, the phase angle master curves exhibited the upside-down bell shape, which means that the phase angle first decreased with the increase in loading frequency, and then increased with the continuous increase in loading frequency. The phase angle master curves showed

notable differences at lower loading frequencies; with the increasing loading frequency, the difference became small, and the curves got together. The phase angle of the PU mixture decrease with the increase in loading frequency or the increase in temperature, which means that the PU mixture exhibited more elastic properties with the increase in loading frequency or the decrease in temperature. This phenomenon consisted of the characteristic of the viscoelastic material, namely that the phase angle would be lower at extremely low temperatures. Therefore, the results of the other four equations except Arrhenius do not comply with the characteristic of the viscoelastic material at a lower temperature, which means that these four shift factor equations were not suitable for Model 4.

For Model 4 with the Arrhenius shift factor equation, different error minimization methods had little effect on the fitting results of the phase angle master curve; all the phase angle master curves fitted by different error minimization methods almost converged together. In addition, the four parameter values ($R^2$, Se/Sy, error$^2$, SSE) fitted by four error minimization methods also had little difference. The $R^2$ maximization method is adopted for comparison in the following discussion, with conclusion the same as the above.

### 4.1.5. Model 5 Fitting Results

For Model 5 in Figure 11, the phase angle master curves with different error minimization methods under the same shift factor equation had similar trend lines, and the phase angle gradually decreased with the increasing loading frequency. At lower loading frequencies, the phase angle master curves showed a considerable difference, and the difference diminished with the increasing loading frequency. The phase angle master curves exhibited a tiny difference when the loading frequency was higher than $10^{-3}$ Hz.

For Model 5, different error minimization methods had limited effects on the phase angle master curves when the loading frequency was higher than $10^{-3}$ Hz. Analyzing the fitting parameter values by five different shift factor equations, the $R^2$ maximization method produced relatively good fitting parameter values compared with the other methods. The $R^2$ maximization method is recommended for equation parameter regression in the following section.

### 4.1.6. Comparison of Different Shift Factor Equations

According to the discussion above, when the loading frequency was higher than $10^{-3}$ Hz, the effect of the error minimization method on the construction of the phase angle master curve was insignificant. The $R^2$ maximization (>0.99) method was recommended as the main constraint for the regression procedure, and the Se/Sy minimization (<0.05), SSE minimization (<0.05), and Error$^2$ minimization (<0.05) were adopted as the additional constraints.

From Figure 12, it can be found that in the same model, the master curves of phase angle with different shift factor equations had a similar trend, but the ultimate values produced by different shift factor equations at lower frequencies had obvious differences. The shift factor equation form had a significant influence on the phase angle master curve of the PU mixture. Then, the difference between measured and predicted dates is compared in the following discussion to recommend the proper shift factor equation for each model.

### 4.2. *Analyzing the Influence of Shift Factor Equation on the Fitting of Phase Angle Master Curve*

The LOE method, the linear fitting method, and the Pearson linear correlation analysis were introduced to evaluate the prediction accuracy of the model used to predict the phase angle from the measured data. For the LOE method, the LOE line was used to determine how closely the predicted data matches LOE and identify and exclude outlier data. The closer the data point is to the LOE, the more accurately the model and shift factor equation fit the data. For the linear fitting method, the parameters including the trend line $R^2$, trend line slope, and residual sum of squares (RSS) were adopted to evaluate the fitting results of different shift factor equations under difficult models. The trend line slope and trend line $R^2$ were generally used as statistics to evaluate the correlation. The linear relationships'

$R^2$ value provides information about how well each model fits the data and how much of the data's variability can be accommodated by that model. Higher $R^2$ and lower RSS values indicate better prediction accuracy. When the trend line slope values were higher than 1, this meant that the prediction procedure would overestimate the measured data; when the trend line slope values were lower than 1, this represented an underestimation of the phase angle. If the trend line slope is closer to 1, the prediction method is more accurate. The bias is minimized when the slope is 1 and the intercept is 0. The model's accuracy was determined by calculating the slope of the linear connection between the predicted and measured values, with the intercept set close to zero. The $R^2$ value of the linear relationship could evaluate the variability of the measured results by each model and indicate the overall accuracy of the model. The closer the $R^2$ value is to 1, the higher the the degree of correlation between the predicted and measured results. The RSS index represents the bias between the predicted and measured results; the smaller the RSS index, the more accurate the prediction model.

The Pearson linear correlation analysis was performed to compare the correlation between the measured and predicted results with a 1% of significance level. If the *p*-value is less than 0.01, the predicted and measured results have a significant correlation, otherwise they are statistically different. A smaller *p*-value indicates a more credible regression equation and a better-fitting effect.

### 4.2.1. Model 1 Fitting Results

From Figure 13, it can be seen that the predicted data spots are almost linear and equally distributed on both sides of the LOE, and the measured and predicted results have a relatively linear association. The prediction accuracy of different shift factor equations had some differences. According to the trend line $R^2$ value and the RSS value in Table 1, the prediction accuracy of different shift factor equations was ranked as Polynomial > WLF > Arrhenius > Log-linear > Kaelble, and the value of the trend line $R^2$ varied from 0.9201 to 0.96584. For Model 1, the Polynomial shift equation had the strongest $R^2$ value and the lowest RSS value and showed the best goodness-of-fit statistics. This means that the Polynomial shift factor equation could provide better prediction accuracy compared with the measured values.

The higher the trend line $R^2$ value, the closer the predicted date spots to the LOE. The shift factor equations with higher trend line $R^2$ values predict more accurate data compared with the measured results. All the trend line slopes were smaller than 1, which means that the shift factor equations combined with Model 1 underestimated the measured results. The higher the trend line $R^2$ value, the closer the trend line slope to 1.

Based on the statistical analysis, compared with the measured data, the *p*-values of all five shift factor equations were smaller than 0.01. These analyzing results indicated that there was a statically acceptable linear association between measured and predicted data at a 99% significance level. The Kaelble shift factor equation had the strongest Pearson linear correlation coefficient, indicating that the predicted data by the Kaelble shift factor equation had the strongest correlation with the measured result.

### 4.2.2. Model 2 Fitting Results

It can be found from Figure 14 that under every shift factor equation, the predicted data spots were orderly distributed along the LOE, and no obvious discrete values were discovered. Based on the trend line $R^2$ and RSS values in Table 2, the prediction precision of different shift factor equations was ranked as Kaelble > Polynomial > WLF > Arrhenius > Log-linear, and the value of the trend line $R^2$ varied from 0.95745 to 0.9789, which was higher than that of Model 1. The RSS values followed the opposite trend and also proved the same rank of different shift factor equations. The Kaeble shift factor equation showed the best goodness-of-fit statistics with the strongest $R^2$ value and the smallest RSS value, which also proved that the Kaelble shift factor equation could provide better accuracy predictions than the other equations.

The trend line slopes of the WLF shift factor equation were bigger than 1, which means that this equation combined with Model 2 would over-estimate the measured data. The data spots produced by the WLF shift factor equation were almost higher than the LOE, which also complied with the findings of the trend line slope. The trend line slopes of the other four shift factor equations were all smaller than 1, which means that those equations would underestimate the measured results.

According to the Pearson linear correlation analysis, the *p*-values of all five shift factor equations were smaller than 0.01. A statically acceptable linear association between measured and predicted data is at a 99% significance level. The Pearson linear correlation coefficient of the Kaeble shift factor equation was the biggest, so the Kaeble shift factor equation had the best prediction accuracy.

### 4.2.3. Model 3 Fitting Results

From Figure 15, it can be seen that the result spots predicted by different shift factor equations under Model 3 were all linearly distributed around the LOE, and the bias of all shift factor equations was not obvious. The prediction accuracy of all shift factor equations under Model 3 was ranked as Kaelble > Polynomial > WLF > Arrhenius > Log-linear according to the trend line $R^2$ values and RSS values in Table 3, and the order was the same as that in Model 2. The trend line $R^2$ values ranged from 0.95847 to 0.9792, which was similar to Model 2. The Kaelble shift factor equation had the highest trend line $R^2$ value and the smallest RSS value. These results suggested that the Kaelble shift factor equation had the best goodness-of-fit statistics and could predict the phase angle more precisely than the other equations under Model 3.

The data predicted by the shift factor equation with the strongest trend line $R^2$ value was closer to the LOE than the other equations. The trend line slope of all shift factor equations also followed the same order as the trend line $R^2$ values. The trend line slopes were all smaller than 1, which means that all the shift factor equations under Model 3 slightly underestimated the measured data.

The *p*-values of all five shift factor equations were all smaller than 0.01 under the Pearson linear correlation analysis, which indicates that the measured and predicted data had a very significant correlation at a 99% significance level. The Kaelble shift factor equation had the highest Pearson linear correlation coefficient, so the Kaelble shift factor equation could produce more precise forecasts than the other equations.

### 4.2.4. Model 4 Fitting Results

From Figure 16, it can be discovered that the data spots predicted by all shift factor equations under Model 4 were scattered along the LOE, and there were no discrete values. According to the trend line $R^2$ values and RSS values in Table 4, the prediction precision of all shift factor equations was as follows: Kaelble > Polynomial > WLF > Log-linear > Arrhenius and the corresponding trend line $R^2$ values increased from 0.96108 to 0.98577. The RSS values also proved the rank order. The Kaelble shift factor equation had the strongest trend line $R^2$ value and the smallest RSS value. The equation could predict more precise results than the other equations. The data spots would be closer to the LOE with the higher trend line $R^2$ value and smaller RSS value. The trend line slope of the Kaelble shift factor equation approached 1 more prominently than the other shift factor equations, and all the trend line slopes of different shift factor equations were slightly smaller than 1, which means that the shift factor equations underestimated the measured phase angle.

The linear association between the measured and predicted data was calculated using the Pearson linear correlation coefficient. According to the SPSS analysis, the *p*-values of all shift factor equations were smaller than 0.01. Therefore, there was a statically acceptable linear association between measured and predicted phase angle for all shift factor equations under Model 4 at a 99% significance level. The Kaelble shift factor equation had the highest Pearson linear correlation coefficient, which also proved that the Kaelble shift factor equation could provide a better prediction more accuracy than the other equations.

4.2.5. Model 5 Fitting Results

From Figure 17, it can be seen that the data spots obtained by the WLF shift factor equation were all above the LOE, which means that the WLF shift factor equation overestimated the measured results. The predicted result spots from the other four shift factor equations did not show obvious discrete values; all the spots were grouped close along the LOE, which indicated good agreement between the measured and predicted values. The prediction precision of all shift factor equations in accord with the trend line $R^2$ values and RSS values in Table 5 ranked as Kaelble > Polynomial > WLF > Arrhenius > Log-linear, which was the same as the results for Models 1 and 2. The trend line $R^2$ values ranged from 0.97729 to 0.95163, and the RSS values also proved the equation order. For Model 5, the Kaelble shift factor equation could provide more precise predictions than the other equation with the best goodness-of-fit statistics.

The higher the trend line $R^2$ values were, the closer the data spots were to the LOE. The trend line describes the relationship between the measured and predicted values. Except the trend line slope of the WLF shift factor equation, which was higher than 1, the slopes of the other four shift factor equations followed the same order as the $R^2$ values. The slopes were slightly lower than 1, which indicated that the four shift factor equations would somewhat underestimate the measured results.

According to the Pearson linear correlation analysis, the *p*-values of all five shift factor equations were smaller than 0.01, which suggested a statically acceptable linear association between measured and predicted data. The Kaelble shift factor equation had the highest Pearson linear correlation coefficient and the strongest correlation with the measured data.

Based on the results discussed above, the Polynomial shift factor equation for Model 1 and the Kaelble model for the other four shift factor equations were recommended for shifting the phase angle at different test temperatures and obtaining the phase angle master curves.

### 4.3. Comparing Master Curves with Recommended Models and Shift Factor Equations

4.3.1. Comparing Polynomial Shift Factor Equation Fitting Results

Under the Polynomial shift factor equation, the data spots predicted by all the models were uniformly distributed along the LOE, and no obvious discrete values were discovered. All the trend line slopes of the five models were smaller than 1, ranging from 0.94916 to 0.97043, so all models combined with the Polynomial shift factor equation would slightly underestimate the measured data. The trend line $R^2$ value of Model 4 was the highest (0.97464), and the corresponding RSS value was the smallest (1.87013), which means that Model 4 had the best fitting result and the smallest errors between the measured and predicted data values. The trend line $R^2$ values of the other four models were close, which ranged from 0.96435 to 0.96584, and the RSS values ranged from 2.50653 to 2.73905, which means that the other four models had similar fitting results, and the errors between the measured and predicted data values also had little difference.

The trend line $R^2$ values of models used for PU mixtures were much higher than those for asphalt mixtures. For example, in literature [56], the trend line $R^2$ values ranged from 0.851 to 0.858 for the SLS, GLS, and SCM models. Therefore, Model 4 exhibited the best-fit statistic, and Model 4 is capable to fit the measured data much more closely in comparison with the other models, which is in agreement with the conclusion of literature [56].

4.3.2. Comparing Kaelble Shift Factor Equation Fitting Results

Under the Kaelble shift factor equation, as shown in Figure 19 and Table 7, it can be concluded that the spots predicted by Model 1 were relatively away from the LOE compared with the other models, the trend line $R^2$ value was the smallest, and the corresponding RSS value was the biggest. This means that Model 1 with Kaelble shift factor equation had the worst fitting results and the lowest prediction precision compared with the other models. Model 4 had the strongest $R^2$ value (0.98577), the highest trend line slope (0.98378), and the smallest RSS value (1.09921), which suggested that Model 4 could have the best-fit statistic

and produced better prediction accuracy than the other models under the Kaelble shift factor equation. The other three models had similar fitting results, e.g., trend line $R^2$ ranged from 0.97729 to 0.97792, trend line slope ranged from 0.97563 to 0.98029, and the RSS values ranged from 1.59565 to 1.75707.

Based on the discussion above, Model 4 combined with the Kaelble shift factor equation showed the best predicting accuracy for the phase angle of PU mixtures combined with the difference in trend line $R^2$, slope, and RSS values. However, the shape of the master curve of Model 4 with Kaelble shift factor did not comply with the changing regularity of the mixture which was explained in Section 4.1.4. Model 3 with Kaelble shift factor equation was recommended for the construction of the phase angle master curve of the PU mixture.

*4.4. Comparing the Master Curves under Different Models*

From the visual inspection of Figure 18, it can be seen that when the loading frequency was smaller than $10^{-3}$ Hz, the phase angle master curve of Model 5 was extremely higher than that of the other models, and the GLS and CAM model had the same trend, while Model 1 had lower values. The master curves of the models excluding Model 4 would decrease monotonously with the increase in the loading frequency as expected, the master curve of Model 4 would first decrease and then increase with the increase in the loading frequency. When the loading frequency was between $10^{-3}$ and $10^3$ Hz, the difference between all the master curves became small. The shape of Model 1 was similar to that of the results of the dense PU mixture in [57].

The phase angle master curve of the PU mixture did not show the traditional smooth "Bell" shape reported in [28] and did not have the highest value at intermediate loading frequency, which was different from the master curve of the phase angle of the asphalt mixture. For the phase angle master curve of the asphalt mixture, the master curves of phase angle initially rise with the increase in loading frequency to a maximum value and then fall when the loading frequency is increased even more. The "Bell" shape of the phase angle master curve can reflect the properties of the asphalt mixture in the full loading frequency range. At low temperatures, the asphalt mixture acts elastically and is mainly subjected to asphalt binder, thus the phase angle is smaller. When decreasing frequency (or increasing temperature), the asphalt binder becomes soft and approaches viscous, then leading to the increment in phase angle. At high temperatures, the asphalt mixture mostly exhibits viscous behavior and the asphalt binder's effect on the mixture weakens while the interlocking force between aggregates strengths, then leading to the decrease in phase angle. Thus, with further increase in temperature (or decrease in frequency), the phenomenon of aggregate skeleton mainly bearing the loading stress becomes more obvious, leading to a smaller phase angle [58].

The PU mixture decreased monotonically with the frequency increased. This phenomenon means that the PU mixture would exhibit viscous property at low loading frequency or high temperature and show the elastic property at high loading frequency or low temperature. For linear viscoelastic material, it is generally expected that the phase angle increases as the loading frequency decreases. The PU mixture's phase angle complied with the regularity of the linear viscoelastic material.

The asphalt component contributes to the mixture's viscoelasticity. Consistent with previous studies, the viscoelastic properties of asphalt affect the phase angle of the asphalt mixture; a strong sinusoidal relationship exists between the phase angle of asphalt and the asphalt mixture [59–62]. When the asphalt binder is no longer the dominant component in the mixture, the aggregates take over and the phase angle master curve reaches its top [63]. Therefore, the PU is the source of the viscoelastic of the PU mixture, and the PU mixture has less temperature susceptibility the asphalt mixture and does not change the behavior transition.

## 5. Conclusions

This paper adopted five master curve models, five shift factor equations, and four error minimization methods for the phase angle of the asphalt mixtures to evaluate the feasibility of constructing the phase angle master curve for the PU mixture. Three steps were performed to evaluate the effectiveness of the error minimization method, shift factor equation, and master curve model sequentially. According to the discussion above, some conclusions could be concluded as follows.

(1) When the loading frequency was higher than $10^{-3}$ Hz, the master curves fitted by different error minimization methods under the same master curve model and shift factor equation exhibited little difference. The $R^2$ error minimization method, including $R^2$ (>0.99) as the main constraint, Se/Sy (<0.05), SSE (<0.05), and Error$^2$ (<0.05) as the additional constraints, was used for parameter regression.

(2) The LOE method, linear fitting method, and Pearson linear correlation analysis proved to be effective methods to evaluate the precise fitting of the phase angle from the test data.

(3) According to the linear fitting and statistics analysis, the Kaelble shift factor equation had the strongest correlation with the measured result for Models 2, 3, 4, and 5. Then, for Model 1, the Polynomial shift factor equation could provide more accurate predictions than the other equations.

(4) The combination of Model 3 and the Kaelble shift factor equation was recommended for fitting the phase angle master curve of the PU mixture.

(5) The trend line $R^2$ values of the linear fitting results for the PU mixtures were much higher than those with the same master curve models for the asphalt mixture, which indicated that the PU mixture had lower temperature sensitivity than the asphalt mixture.

(6) For all the phase angle master curves, Model 5 had the highest values, Models 2 and 3 had the same trend, Model 1 had the lowest value, and all four models decreased monotonously with the increase in the loading frequency. Model 4 first decreased and then increased with the increase in the loading frequency. When the loading frequency ranged between $10^{-3}$ and $10^3$ Hz, all the master curve models exhibited insignificant differences.

(7) The phase angle master curve did not show the "Bell" shape as that of the asphalt mixture and did not have the highest phase angle values at intermediate loading frequency. For the asphalt mixture, at higher loading frequency or lower temperature, the asphalt mixture exhibited elastic properties and was mainly subject to the asphalt. With the increase in temperature or the decrease in loading frequency, the asphalt became soft, and the asphalt mixture exhibited viscosity and was subject to the aggregate skeleton. For the PU mixture, the viscous property component increased with the increase in temperature or the decrease in loading frequency, but the PU mixture was still subject to the PU, which was different from that of the asphalt mixture. The phase angle behavior of the PU mixture complied more with the characteristic of the viscoelastic material compared with the asphalt mixture.

In this paper, the compared five master models and five shift factor equations exhibited significant differences when the loading frequency was lower than $10^{-3}$ Hz. Therefore, the phase angle master curve models introduced from the asphalt mixture were only suitable for the PU mixture when the loading frequency was higher than $10^{-3}$ Hz. There are few studies conducted about the phase angle master curve of the PU mixture; this paper analyzed the influence factors during the fitting process of the phase angle of the PU mixture and contributed to the construction of the phase angle master curve and prediction of the characteristic of the PU mixture at extreme temperatures or loading frequencies, which could be used to predict the road performance of the PU mixture and determine the proper pavement structure combined with the PU mixture layer.

More attention is needed for studying the phase angle master models and shift factor equations to extrapolate to frequency ranges lower than $10^{-3}$ Hz. More phase angle

master curves and shift factor equations should be compared in the future for a better understanding of the viscoelastic property of the PU mixture.

**Author Contributions:** Conceptualization, H.Z.; methodology, S.M.; software, P.Z., validation, H.Z., X.W. (Xiaoyan Wang) and J.W.; formal analysis, H.Z.; investigation, H.Z. and S.C.; resources, X.W. (Xiufen Wang), B.J. and W.Z.; data curation, H.Z., S.C., X.W. (Xiaoyan Wang) and S.L.; writing— original draft preparation, H.Z.; writing—review and editing, S.M.; visualization, H.Z., S.C., P.Z. and J.W.; supervision, S.M.; project administration, X.W. (Xiufen Wang), W.Z. and S.M.; funding acquisition, W.Z. All authors have read and agreed to the published version of the manuscript.

**Funding:** This research received no external funding.

**Institutional Review Board Statement:** Not applicable.

**Informed Consent Statement:** Not applicable.

**Data Availability Statement:** The data presented in this study are available on request from the corresponding author.

**Acknowledgments:** We thank Guang Li, and Fuxiu Liu for their assistance with experiments and valuable discussion.

**Conflicts of Interest:** The authors declare no conflict of interest.

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
