# Peer review of "Study on the Phase Angle Master Curve of the Polyurethane Mixture with Dense Gradation"

_coatings, doi:10.3390/coatings13050909_

Round 1

Reviewer 1 Report

Dear Authors,

The article presents interesting research results. I propose the following modifications.

1. In the introduction, it is worth referring to similar materials used in modern 3D printing technologies:

- Viscoelastic Properties of Cell Structures Manufactured Using a Photo-Curable Additive Technology-PJM, DOI10.3390/polym13111895

2. Suggests in point 2.1 Material to list how many samples for each test variant have been made. The number of test samples is a key issue to ensure the credibility of the research.

3. In the abstract, it is suggested not to use the abbreviations CAM, but to give the full name and explain all abbreviations in the text.

4. The article should be corrected in terms of the editorial requirements of the journal.

5. Please explain in more detail what the differences in tables 1 and 2 for SSR and other functions result from.

6. It proposes to extend the description of the point (4.1.4 Modified CAM model fitting results) with an in-depth analysis of the results in terms of fitting.

7. Please consider condensing research conclusions into smaller ones, but I do not require this as a critical remark.

It is Minor Review.

Kind regards,

Reviewer

Author Response

I appreciate your suggestions and valuable advice. Based on the suggestions, I modified the manuscripts as follows.

Point 1: In the introduction, it is worth referring to similar materials used in modern 3D printing technologies:

- Viscoelastic Properties of Cell Structures Manufactured Using a Photo-Curable Additive Technology-PJM, DOI10.3390/polym13111895

Response 1: Thank you for the supplied valuable references. The useful information from these two references was presented as reference 41 in the renewed manuscript.

Point 2:  Suggests in point 2.1 Material to list how many samples for each test variant have been made. The number of test samples is a key issue to ensure the credibility of the research.

Response 2: The average dynamic modulus values of two replicates were used for the construction of the master curve. More information about the test was added in section 2.1 in the new manuscript.

Point 3:  In the abstract, it is suggested not to use the abbreviations CAM, but to give the full name and explain all abbreviations in the text.

Response 3: Thank you for reminding the mistakes, I have checked the manuscript about the abbreviations, and the full name and explanation were added in the new manuscript.

Point 4:  The article should be corrected in terms of the editorial requirements of the journal.

Response 4: The manuscript was rewritten with the editorial requirements of the journal, and all the mistakes were corrected.

Point 5:  Please explain in more detail what the differences in tables 1 and 2 for SSR and other functions result from.

Response 5: More explanations about the data in Tables 1 and 2 were added to the new manuscript.

Point 6:  It proposes to extend the description of the point (4.1.4 Modified CAM model fitting results) with an in-depth analysis of the results in terms of fitting.

Response 6: Thank you for reminding me. More explanation and description were added in the new manuscript.

Point 7:  Please consider condensing research conclusions into smaller ones, but I do not require this as a critical remark.

Response 7: Thank you for the advice. The conclusion section was improved in the new manuscript.

Reviewer 2 Report

The findings of this study suggest that the compared 5 master models and shift factor equations show significant differences at loading frequencies lower than 10^-3 Hz. As a result, the phase angle master curve models derived from asphalt mixture data are only applicable to the PU mixture at higher loading frequencies above 10^-3 Hz.

The grammar should be improved in the text, there are some typos and inaccuracies, for example:

Lines 113-114 “axe” instead of “axis”

Line 314 “maser” instead of “master” 

 Line 761 "IWe" etc.

The description of the calculated dependencies should be given in more detail. In my opinion, it is much more convenient when the discussion of the results is indicated immediately after the experimental data, and not in a separate section. It was very difficult to match the graphics with the text.

I have found only some typos. English is ok.

Author Response

I appreciate your suggestions and valuable advice. Based on the suggestions, I modified the manuscripts as follows.

Point 1:  Lines 113-114 “axe” instead of “axis”, Line 314 “maser” instead of “master”,  Line 761 "IWe" etc.

Response 1: Thank you for the reminder, all the mistakes are corrected.

Point 2:  The description of the calculated dependencies should be given in more detail. In my opinion, it is much more convenient when the discussion of the results is indicated immediately after the experimental data, and not in a separate section. It was very difficult to match the graphics with the text.

Response 2: More descriptions were added in the new manuscript about the calculated dependencies.

The sections of test results and discussion were in separate sections, this is the template requirements of the journal provided. I am sorry about the confusion.

Reviewer 3 Report

Memorandum

Subject: Review, April 17, 2023

Coatings

Study on the Phase Angle Master Curve of the Polyurethane Mixture with Dense Gradation

Haisheng Zhao 1, 2, Xiufen Wang 3 , Shiping Cui 1 , Bin Jiang 4 , Shijie Ma 1, *, Wensheng Zhang 4 , Peiyu Zhang 1 , 4 Xiaoyan Wang 1 , Jincheng Wei 1 , Shan Liu 1

Comments:

1.       The authors must and should consider adding a nomenclature to identify parameters and abbreviations used throughout the paper.

  1. The abstract is hard to follow and unclear, the authors should re-write it and focus on what is being presented in the paper along with a sentence citing the outcome of the work.

3.       Most of figures are missing labels on both X and Y axis.

4.       The authors approach to the analytical calculations is not very clear, no clear discussion outlining the work to be performed is present either. It is not well defined and obvious. The authors should elaborate on this by adding additional details with some explanation citing what is being done.  For instance, was there any experimental work performed ??....or only analytical….!!!??? This need to be indicated and explained.

5.       The analysis is mainly focused on presenting series of graphical data throughout the paper without providing a notable link between the analysis performed.  This makes it difficult to comprehend what is being exhibited. In another word, the analytical modeling is missing, it does not indicate how it was done and what tools were used to run the analysis.

6.       For example, many of the models such as the Modified CAM in Figure 4 lacks the empirical relationship derived from the curve fitting. Many more are equally missing in other plots.

7.       The authors refer to predicted data versus measured data without indication what was measured and how it was acquired.

8.       The authors in Figure1 refer the Gradation of PU mixture, this does not indicate how this data was generated.

9.       The paper is too long, it further lacks consistency and clear delivery of the work/results accomplished.

10.   The conclusion is very long and does not offer a clear concluding remark. The authors should cite what was accomplished and what may have impacted the outcome of the study if any exists. Considering a bullet type statement citing what was found and if anything may have impacted the results would serve the reader better.

Overall, the paper is lengthy and needs major work. The authors should focus on presenting the key findings that are relevant to their effort. It also needs some reorganization and some major editorial work up. I cannot recommend it for publication in its present form. The authors would further benefit from reducing the size of the article to a more concise format that offers meaningful presentation of their work.

Recommend professional review

Author Response

I appreciate your suggestions and valuable advice. Based on the suggestions, I modified the manuscripts as follows.

Point 1: The authors must and should consider adding a nomenclature to identify parameters and abbreviations used throughout the paper.

Response 1: Thank you for the suggestion, I will try my best to add nomenclature for clear reading.

Point 2:  The abstract is hard to follow and unclear, the authors should re-write it and focus on what is being presented in the paper along with a sentence citing the outcome of the work.

Response 2: Thank you for your advice, the abstract is rewritten and the outcome of this paper would also be added.

Point 3:  Most of figures are missing labels on both X and Y axis.

Response 3: Thank you for the reminder, the missing labels were added in the new manuscript.

Point 4: The authors approach to the analytical calculations is not very clear, no clear discussion outlining the work to be performed is present either. It is not well defined and obvious. The authors should elaborate on this by adding additional details with some explanation citing what is being done.  For instance, was there any experimental work performed ??....or only analytical….!!!??? This need to be indicated and explained.

Response 4: Thank you for your advice, more explanation about the previous studies and conclusion were added in the new manuscript.

Point 5:  The analysis is mainly focused on presenting series of graphical data throughout the paper without providing a notable link between the analysis performed.  This makes it difficult to comprehend what is being exhibited. In another word, the analytical modeling is missing, it does not indicate how it was done and what tools were used to run the analysis.

Response 5: I am sorry about the confusion in this paper. A new section 2.4 fitting process was added into the new manuscript to explain the detailed fitting and analyzing procedure of this paper, the methods and how the equations were used were explained in this section.

Point 6:  For example, many of the models such as the Modified CAM in Figure 4 lacks the empirical relationship derived from the curve fitting. Many more are equally missing in other plots.

Response 6: The Figures 2−6 represent the models at specific status, e.g., all parameters equal to 1 or 0. These figures were used to show the typical curves of different models and also show the effect of different parameters on the shape of different models, some influence the width, and some affect the height of the shape of the model.

Point 7:  The authors refer to predicted data versus measured data without indication what was measured and how it was acquired.

Response 7: More explanation and description were added to section 2.1 in the new manuscript.

Point 8:  The authors in Figure1 refer the Gradation of PU mixture, this does not indicate how this data was generated.

Response 8: More explanation and description were added to section 2.1 in the new manuscript.

Point 9:  The paper is too long, it further lacks consistency and clear delivery of the work/results accomplished.

Response 9: There are many pictures in this paper which increase the pages of this paper. This paper followed the main line, firstly, the effect of four error minimization methods on the fitting results under different master curve models and shift factor equations would be compared and analyzed, then the proper error minimization methods would be recommended for the master curve fitting procedure. Secondly, after the error minimization method was determined, the effect of five different shift factor equations on the phase angle master curve fitting under different master curve models was compared and analyzed, then the proper shift factor equations with the best fitting accuracy would be recommended for the phase angle master curve fitting procedure. Finally, after the error minimization method and shift factor equation were determined, the effect of five different master curve models on the phase angle master curve fitting was compared and analyzed, then the most proper master curve models would be recommended.

This paper was organized in the progressive mode to determine the proper error minimization method, shift factor equation, and master curve model for the phase angle master curve construction of the PU mixture. Because there are few studies on the phase angle master curve construction of the PU mixture, this paper needs to study the influence factors that could affect the phase angle master curve construction, e.g., error minimization method, shift factor equation, and master curve model, to provide some contribution for the study of the PU mixture.

Point 10:  The conclusion is very long and does not offer a clear concluding remark. The authors should cite what was accomplished and what may have impacted the outcome of the study if any exists. Considering a bullet type statement citing what was found and if anything may have impacted the results would serve the reader better.

Response 10: Thank you for the advice. The conclusion section was improved in the new manuscript.

Round 2

Reviewer 3 Report

Comments:

  1. The authors stated  that they have added  a nomenclature to identify the parameters and variables used. No nomenclature is present in the revised version.
  2. The abstract remains unclear, it does not deliver a clear message on what is being presented in the paper along with a sentence citing the outcome of the work.
  3. Figures still do not have labels on both X and Y axis. As an example, figures 2 and 3 and so on…
  4. The paper is much improved; however, it  remains too long. It would serve the reader better if it can be reduced and confined to focusing on the methodology applied and the result obtained.
  5. The conclusion is relatively better, it can be much improved by simplifying the outcome with shorter sentences.
  6. Overall, the paper is reasonably better, however, there is still more work to be done. Addressing the above issues should help improving its scientific value. Additionally, the paper needs major reorganization and editorial workup.

Upon making the above revisions, it can be released for publication.

an editorial workup and reorganization will improve the paper content and add more scientific Value. 

Author Response

Thank you for your careful review and detailed suggestions.

Point 1: The authors stated that they have added a nomenclature to identify the parameters and variables used. No nomenclature is present in the revised version.

Response 1: There are five master curve models and shift factor equations used for the construction of the phase angle master curve, and all the models and equations were widely used for fitting the phase angle of the asphalt mixture, the parameters and variables were commonly accepted, for example, the Standard logistic Sigmoid model was commonly named as SLS model, the β and γ are the shape parameters of the model curve; the Christensen Anderson and Marasteanu model was named as CAM model, and the v, and tc are the shape parameters, but these shape parameters didn’t have the same influence on the shape of different models. Therefore, in this paper, all the parameters were kept in the original form for better understanding. In the revised version, the five models were represented in models 1 to 5 for clear presentation. All the other parameters and variables used were not shown in the nomenclature form.

Point 2: The abstract remains unclear, it does not deliver a clear message on what is being presented in the paper along with a sentence citing the outcome of the work.

Response 2: The last two sentences were rewritten, and the last sentence presents the outcome of this paper.

Point 3: Figures still do not have labels on both X and Y axis. As an example, figures 2 and 3 and so on…

Response 3: Thank you again for the reminder. The explanation about the label on both the X and Y axis were added in lines 163−165.

Point 4: The paper is much improved; however, it remains too long. It would serve the reader better if it can be reduced and confined to focusing on the methodology applied and the result obtained.

Response 4: In this paper, many pictures about the fitting results were provided for better comparing the influence of the error minimization method, shift factor equation, and master curve model. This paper analyzed the phase angle master curve of the PU mixture for the first time, the influence of different factors should be analyzed thoroughly. In this version, pictures were deleted in sections 3.2.1 to 3.2.5 and the data was shown in Table form for reducing the pages. The new version was shortened by 3 pages.

Point 5: The conclusion is relatively better, it can be much improved by simplifying the outcome with shorter sentences.

Response 5: The conclusion was furthermore improved in this version for some sentences.

Point 6: Overall, the paper is reasonably better, however, there is still more work to be done. Addressing the above issues should help improve its scientific value. Additionally, the paper needs major reorganization and editorial workup.

Response 6: This paper was organized to analyze and compare the influence of the error minimization method, shift factor equation, and master curve model on the construction of the phase angle master curve of the PU mixture. Therefore, the structure of this paper was not changed in this version. Some more mistakes were corrected.